

# Arable soil formation and erosion: a hillslope-based cosmogenic-nuclide study in the United Kingdom

Daniel L. Evans[1], John N. Quinton[1], Andrew M. Tye[2], Ángel Rodés[3], Jessica A.C. Davies[1], and Simon.
5   M. Mudd[4]

[1] Lancaster Environment Centre, Lancaster University, Lancaster, Lancashire, UK
[2] British Geological Survey, Keyworth, Nottinghamshire, UK
10   [3] Scottish Universities Environmental Research Centre, East Kilbride, UK
[4] School of GeoSciences, University of Edinburgh, Edinburgh, UK

*Correspondence to*: D. L. Evans (d.evans3@lancaster.ac.uk)

30



## Abstract

Arable soils are critical resources that support multiple ecosystem services. They are frequently threatened, however, by accelerated erosion. Subsequently, policy to ensure their long-term security is an urgent societal priority. Although long-term security relies upon a balance between the rates of soil loss and formation, there have been few investigations of the formation rates of soils supporting arable agriculture. This paper addresses this knowledge gap by presenting the first

10    isotopically-constrained soil formation rates for an arable (Nottinghamshire, UK) and coniferous woodland hillslope (Shropshire, UK). Rates ranged from 0.023 mm year$^{-1}$ to 0.064 mm year$^{-1}$ across the two sites. These rates fall within the range of previously published rates for soils in temperate climates and on sandstone lithologies but significantly differed to those measured in the only other UK-based study. We suggest this is due to the parent material at our sites being more susceptible to weathering. Furthermore, soil formation rates were found to be greatest for aeolian-derived sandstone when

15    compared with fluvially-derived lithology raising questions about the extent to which the petrographic composition of the parent material governs rates of soil formation. On the hillslope currently supporting arable agriculture, we utilised cosmogenically-derived rates of soil formation and erosion in a first-order lifespan model and found, in a worst-case scenario, that the backslope A horizon could be eroded in 137 years with bedrock exposure occurring in 209 years under the current management regime. These findings represent the first quantitative estimate of cultivated soil lifespans in the UK.

**Copyright statement.**

30



# 1 Introduction

Soil erosion is a significant threat to society (Pimentel et al., 1995; UNCCD, 2017). Whilst uncultivated 'pristine' soils may develop steady-state thicknesses, where erosion and production are in dynamic equilibrium (Phillips, 2010), human-induced erosion has led to soil thinning across many landscapes (Montgomery, 2007). Soil erosion, left unchecked, can ultimately lead to the removal of the soil cover and the exposure of the underlying parent material (Amundson et al., 2015). The

development of soil conservation strategies has long been an active field for research and practice (Panagos et al., 2016; Govers et al., 2017). Given any long-term strategy to preserve soil resources relies upon a balance between the rates of soil loss and soil renewal (Hancock et al., 2015), the measurement of soil formation is a fundamental component in these conservation efforts.

The mechanisms associated with soil formation have been studied for over a century, with a focus on the development of soil

horizons and the evolution of soil properties (Dokuchaev, 1879; Jenny, 1941; Bryan and Teakle, 1949; Tugel et al., 2005). Efforts to quantify the rates at which soils form from parent materials have included studying how soil properties change across chronosequences (Turner et al., 2018), developing chemical weathering models (Burke et al., 2007) and, in particular, employing terrestrial cosmogenic radionuclide analysis (Heimsath et al., 1997). In the latter, the concentrations of radioactive isotopes in the bedrock, which are partly dependent upon the rate at which bedrock transforms into soil, are

measured.

Despite the recent advancements in cosmogenic radionuclide analysis, their application in soil science has, arguably, not been fully realized. Moreover, there are three research challenges that may explain this. First, there is a dearth of soil formation rate data. Whilst there have been many attempts at calculating a global average soil formation rate from collating multiple inventories (Alexander, 1988; Montgomery, 2007; Stockmann et al., 2014; Minasny et al., 2015), these datasets

often omit more than 100 countries, particularly in Africa and Europe, presenting a clear rationale for more studies to take place in these areas of the world. Second, over 80% of the soil formation rate inventory, comprising data from Montgomery (2007), Portenga and Bierman (2011) and Stockmann et al. (2014), is attributed to samples taken from outcrops and stream sediments procured from drainage basins. Moreover, only 252 [10]Be-derived rates from this inventory of 1850 stem from samples extracted from underneath the soil mantle. In addition, the majority of these stem from mountain regions and deserts

(Heimsath et al., 1997; Wilkinson et al., 2005; Zhao et al., 2018; Struck et al., 2018). This is partly because the observation and estimation of bedrock weathering rates is most commonly carried out by the geomorphological community, principally to identify the mechanisms behind long-term landscape evolution (Heimsath, 2006; Heimsath and Burke, 2013; Ackerer et

al., 2016; Zhao et al., 2018). As a result, there has been no investment in deriving rates of soil formation for soils that support arable agriculture (Heimsath, 2014), despite these soils being identified as a societal priority (FAO, 2015). Such soils are critical to the delivery of multiple ecosystem services and, for many countries, are one of the most critical resources in ensuring the health of society and sustained economic growth. They are also often intensely managed and thus the loci for

accelerated erosion (Quinton et al., 2010; Borrelli et al., 2017). However, in the absence of soil formation rate data, the magnitude of the threat erosion places on the sustainability of soils and arable production is unknown, amounting to a critical knowledge gap. Third, although the distributions of inventoried soil erosion and formation rates are often presented together to demonstrate the severity of soil erosion (Montgomery, 2007; Minasny et al., 2015), the spread of globally-compiled data is such that it cannot offer a useful forecast of the sustainability of soil at a site scale. Both distributions are platykurtic and

there is substantial overlap in these rates: 0—28.8 mm year$^{-1}$ for soil formation (Minasny et al., 2015) and 0—52.9 mm year$^{-1}$ for soil erosion (Montgomery, 2007). For a greater understanding into the sustainability of soil resources at the local scale, we argue that soil scientists should undertake empirical measurements of both soil formation and erosion in parallel.

In this UK-based study, we present $^{10}$Be-derived soil formation rates for two catena sequences in an arable and coniferous woodland setting. The former are the first of their kind globally and the latter are the first of their kind in Europe. We place

our results in the context of the rates previously derived in similar climatic and petrographic settings around the world. Finally, using previously measured soil erosion rates at the arable site, we calculate first-order soil productive lifespans to infer the long-term sustainability of the soil resource.

## 2.0 Materials and Methods

### 2.1 Site Description

This study measures soil formation down two catena sequences (Figure 1). The first is an arable hillslope at Rufford Forest Farm (RFF), east of Mansfield in Nottinghamshire, UK (53°7'13.43" N, 1°4'39.61" W). The second is a woodland hillslope at Comer Wood (CW), north of Quatford in Shropshire, UK (52°30'30.43" N, 2°22'45.68" W). RFF was selected as it is the site of previous tillage and water-based erosion studies (Quine and Walling, 1991; Walling and Quine, 1991; Govers et al., 1996). Electing CW as a sister site is justified based on its similarities in parent geology, macroclimate and soil physical

properties with RFF as detailed below. A Trimble S6 Total Station was used to measure the relative elevation and slope of the catenas at both sites (Figure 1b).

A reconnaissance study of the parent materials and their feasibility for cosmogenic radionuclide analysis was undertaken in spring 2017. Both sites are underlain by Triassic sandstone. In RFF, the Sherwood sandstone (Chester formation; Olenekian,

247—251 Ma) is described as pinkish to red, medium to coarse grained, pebbly, cross-bedded, and friable. In CW, the New Red sandstone (Bridgnorth formation; Cisuralian, 273—299 Ma) is described as brick-red, soft to medium grained, cross-




bedded and aeolian based. Both RFF and CW sit in a temperate oceanic climate (Cfb), between 96—99 m a.s.l. and 50—71 m a.s.l., respectively. The mean annual precipitation and temperature is 709 mm and 9.8°C at RFF and 668 mm and 9.9°C at CW, respectively (Met Office, 2018).

Both sites are positioned beyond the areal limits of the Late Devensian ice sheet, but studies conducted on similar formations of Triassic Sherwood Sandstone nearby suggest that the weathering of the parent material was partly induced by freeze-thaw processes associated with periglacial active layer development possibly during this period (Tye et al., 2012). Although proglacial glaciogenic deposits have been found in the vicinity of CW, the prevalence of similar deposits on the study hillslope has not been studied. However, unpublished work conducted by the authors suggests that the upper (3—5 m) of the
lithosphere at both sites was subject to high-magnitude sediment transport at least 200,000 BP or before, potentially during the Anglian glaciation (~450,000 BP). The complex land-use and vegetation change in the Sherwood Sandstone outcrop, within which RFF is based, has been extensively studied and mapped by Tye et al. (2013). Following the onset of the Holocene, the area has been dominated by a complex sequence of land-use change including broadleaf woodland (6000— 2000 BC), heathland (43—409 AD) and landscaped heathland for hunting (1600 AD). From at least 1855 AD, RFF has been
under an arable regime and in the last twelve years, the dominant crops have been Winter Wheat and Rye. CW is understood to have been an open field until 1903—1926 and then heathland until 1954. Between 1954 and the present day, however, the site has been continuously occupied by coniferous forest (Evans, 2018).

The soils at RFF are classified as Arenosols (FAO WRB) with weak horizonisation. An Ap loamy-sand horizon (82% sand,
16% silt, 2% clay) thickens from 30 to 75 cm and increases in LOI content from 3.65 to 3.91% from summit to toeslope, respectively. This Ap horizon is underlain by a 5 cm fluvial pebble-bed, typical of the Bunter pebble-beds found in the vicinity (Ambrose et al., 2014). An undifferentiated, weakly-consolidated subsoil steadily grades into saprolitic, moderately-consolidated sandstone. The soils at CW are classified as Cambisols (FAO WRB). Similar to RFF, there is little evidence for horizonisation down the profile at CW. A thin (<5 cm) LFH layer overlays an undifferentiated, weakly-consolidated, sandy
subsoil (94% sand, 5% silt, 1% clay) and grades into moderately-consolidated saprolitic sandstone.




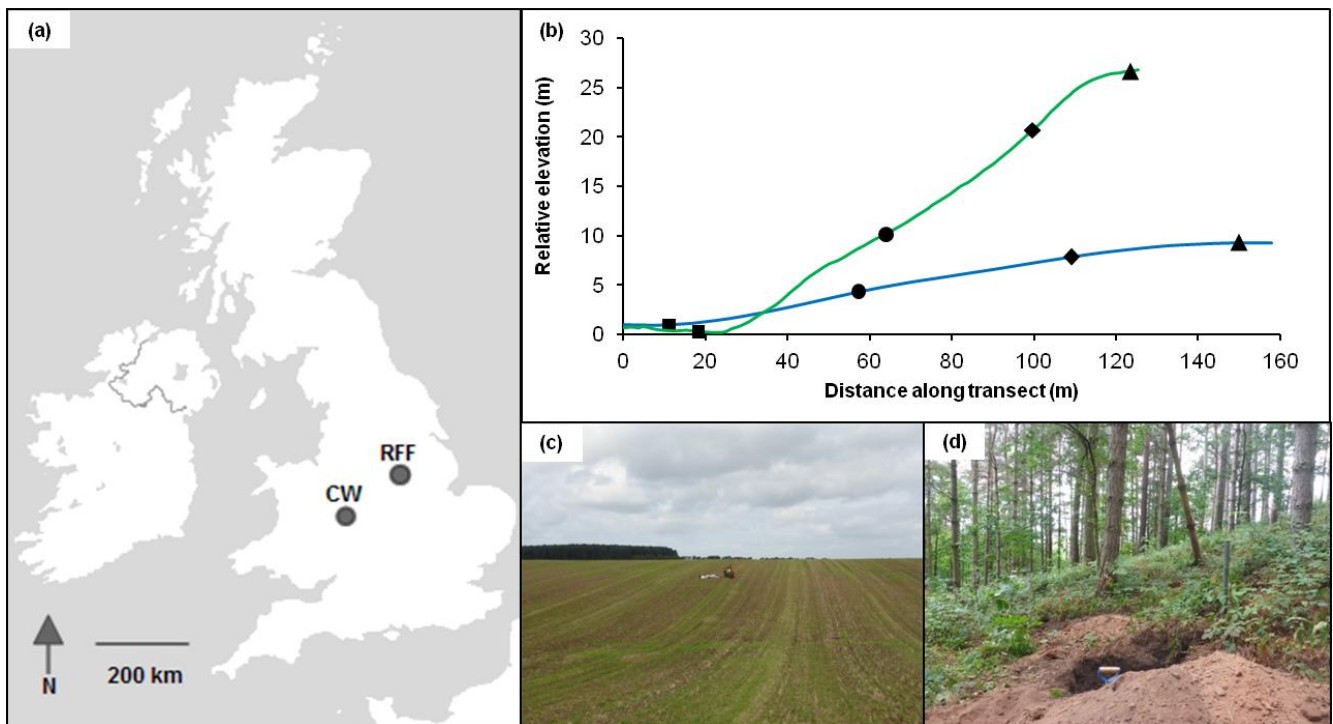

**Figure 1: Locations of the study sites in this paper (a) with elevation profiles (b) for both Comer Woodland (CW; green) and Rufford Forest Farm (RFF; blue). The position of summit (triangles), shoulder (diamonds), backslope (circles) and toeslope (squares) sampling positions are indicated on each profile. Photographs of RFF (c) and CW (d) were taken by the author at the time of sampling.**



## 2.2 Saprolite Extraction and Processing

Four positions (summit, shoulder, backslope and toeslope) along a catena transect were selected for depth to bedrock surveys and saprolite extraction. First, a dynamic cone penetrometer was used to estimate the depth of the soil-saprolite interface. At RFF, a percussion drilling rig then proceeded to extract a series of undisturbed core samples of the soil and saprolite. Cores

were later halved lengthways and, using both the penetrometer data and observations on the competency of the extracted material, the soil-saprolite interface was demarcated. Two samples of saprolite (5 cm thickness) were then subsampled for cosmogenic radionuclide analysis; one at this interface and one from 50 cm below. At CW, following the use of the dynamic cone penetrometer to locate suitable sites, a soil pit was manually dug at each of the four sampling locations. The data derived from the penetrometer and observations of differentiating competency down the profile wall were used to ascertain

the position of the soil-saprolite interface. A sample of saprolite (5 cm thickness) was then extracted from this interface for cosmogenic isotope analysis.

The bombardment of quartz minerals in the uppermost metres of bedrock with cosmic rays leads to the production of [10]Be. Assuming the intensity of these cosmic rays and the in situ weathering of bedrock (ε) is constant, the concentration of [10]Be (N) in a sample of bedrock, Eq. (1), is dependent upon the balance of two factors: the time that the bedrock has been exposed

to cosmic rays with longer durations leading to greater concentrations and the weathering of this bedrock into mobile regolith (soil) with greater rates of bedrock weathering leading to smaller concentrations (Lal, 1991; Stockmann et al., 2014):

$$N = \sum_{i=sp,\mu_f,\mu^-} \frac{P_i(\theta) \cdot e^{-z\rho/\Lambda_i}}{\lambda + \epsilon\rho/\Lambda_i} \qquad (1)$$

where: P are the annual production rates of [10]Be by spallation, fast muons and stopping muons (sp, μf and μ-) at a surface with slope Θ; z is the sample depth; p is the mean density of parent material; λ is the decay constant of [10]Be with λ equalling

In2/[10]Be half-life; and Λ are the mean attenuation of cosmic radiations (Lal, 1991). Production rates, decay constants and attenuation lengths were calculated using field data and the CRONUS-Earth online calculator v2.3 Matlab code for the St scheme (Balco, 2008). As N can be measured using Accelerator Mass Spectrometry (AMS), Eq. (1) can be solved for ε by simple interpolation of N.

A total of twelve samples of saprolite (eight from RFF and four from CW) were prepared for AMS at the Cosmogenic

Isotope Analysis Facility, East Kilbride, Scotland. This comprised of mineral separation, quartz cleaning and procedures leading to the preparation of BeO sample cathodes (Kohl and Nishiizumi, 1992; Fifield, 1999; Corbett et al., 2016). The AMS measurements were carried out at the SUERC AMS laboratory (Xu et al., 2010). [10]Be concentrations are based on 2.79 x10[-11] [10]Be/[9]Be ratio for the NIST Standard Reference Material 4325. The processed blank ratio ranged between 6 and 13%





of the sample $^{10}Be/^9Be$ ratios. The uncertainty of this correction is included in the stated standard uncertainties. Concentrations of $^{10}Be$ were subsequently determined, following Balco (2006) (see Supplementary Table 1).

The local annual production rate of $^{10}Be$ at each study site must also account for any obstructions that reduce the cosmic ray flux to the parent material (Phillips et al., 2016). For an obstruction to cause this reduction, it is required to be several metres

thick which equates, in practice, to topographic features at the scale of tens of meters or greater. The shielding factor, therefore, is a ratio of the $^{10}Be$ production rate at the obstructed site to that at an identical site but with a flat surface and a clear horizon (Balco, 2008). To calculate both shielding factors and subsequently normalize local $^{10}Be$ production rates, site elevation, latitude and longitude were inputted into the CRONUS-Earth Matlab code v2.3 using Lal/Stone (St) scaling (Balco, 2008).

## 2.3 Lifespan analysis at Rufford Forest Farm

To provide an insight into the sustainability of the soil profiles at RFF under arable agriculture, in terms of the balance of erosion and formation, a first-order lifespan model was employed. Calculating the sustainability of a net-eroding soil in first-order terms has been attempted in the past (Elwell and Stocking, 1984; Sparovek and Schnug, 2001; Montgomery, 2007;

Medeiros et al. 2016). Early models (Stocking and Pain, 1983), however, did not account for mass inputs into the soil system, such as that derived from bedrock weathering. In this study, this omission was addressed by using soil formation rates empirically measured at RFF. Furthermore, in previous models, the solum thickness used to calculate the soil lifespan is not universally consistent. Some authors constrain the lifespan by the minimum depth required for primary production (Stocking and Pain, 1983; Elwell and Stocking, 1984). Notwithstanding the fact that this soil threshold depth will, in part, be

crop-dependent, soils that fall below this threshold may still be able to fulfil some of the ecosystem services, such as the sequestration of carbon. To address this here, two lifespan (L) scenarios were calculated, both of which are based on the continuation of contemporary arable agriculture. The first referred to the expected lifespan of the current A horizon (D = 30 cm across the catena). Here, we did not account for any transformation of subsoil into topsoil, which could occur if erosion rates are sufficiently low. The second estimated the time until the underlying parent material is exposed. Here, the observed

depth to bedrock at each catena position was employed.

Both lifespan scenarios were calculated for summit, shoulder, backslope and toeslope catena positions. Three different erosion rates (E) were applied. First, a mean annual erosion rate of 1.19 mm year$^{-1}$ was used based on 137Cs-based data (n = 103) measured by Quine and Walling (1991) at RFF. This mean value represents all erosion processes, including water-based and tillage-based erosion. Two additional lifespans were calculated using rates from the 5th and 95th percentiles of

this dataset (0.19 mm year$^{-1}$ and 2.2 mm year$^{-1}$, respectively).





The soil formation rates, as empirically measured in this paper, were then plotted to derive the soil production function $P$; Eq. (2):

$$P = W\,e\left(\frac{-h}{\gamma}\right) \tag{2}$$

where W is the production rate at zero soil thickness (h) and $\gamma$ is a parameter that determines the thickness of soil when soil formation falls off by 1/e. In this study, $\gamma$ was calculated as being 2.25 m, which is substantially greater than that previously reported (e.g. Heimsath, 1997). It was therefore concluded that soil formation rates at RFF are relatively insensitive to changes in soil thickness. As a result, constant soil formation rates (F) for each catena position, together with two additional rates representing upper and lower standard deviations, were used to calculate soil lifespans. Furthermore, the expected increase in soil formation rates as a result of soil thinning were captured within these upper and lower uncertainties. Soil lifespans were thus calculated using Eq. (3):

$$L = \frac{D}{E - F} \tag{3}$$

where D is depth in mm, E is gross annual soil erosion rate in mm year[-1] and F is gross annual soil formation rate in mm year[-1].

## 3.0 Results and Discussion

### 3.1 Soil Formation Rates

Soil formation rates calculated from measured [10]Be concentrations at RFF range from 0.023 ± 0.002 mm year[-1] to 0.051 ± 0.002 mm year[-1], with the mean soil formation rate being 0.037 ± 0.003 mm year[-1] (Table 1). At CW, soil formation rates range from 0.034 ± 0.001 mm year[-1] to 0.064 ± 0.004 mm year[-1], with the mean soil formation rate being 0.046 ± 0.007 mm year[-1], which is 0.009 mm year[-1] greater than that at RFF. These rates indicate declining soil formation rates with increasing soil thickness (Fig. 2—3). In accordance with geomorphological theory (Conacher and Dalrymple, 1977; King et al., 1983; Pennock, 2003; Schaetzl, 2013), soils are thinner on the slope convexities and the steepest gradients where surface erosion is considered most prevalent. In contrast, soil thicknesses are greater at the summit where surface erosion has been less extensive and the toeslope zone where sediment is deposited. In RFF, formation rates are 0.018 mm year[-1] faster for shoulder and backslope positions where soils are thinner. These results are consistent with many theorized mechanisms that



demonstrate how parent material overlain by shallower soils is more affected by diurnal thermal stresses, contact with water and physical disturbance which can together proliferate physical and chemical weathering processes and thus the conversion of saprolite into soil. Conversely, it was found that formation rates are slower at summit and toeslope positions where the increasing thickness of the soil mantle buffers the parent material from any subaerial factors that may otherwise proliferate

5    weathering (Carson and Kirby, 1972; Cox et al., 1980; Dietrich et al., 1995; Minasny and McBratney, 1999; Wilkinson and Humphreys, 2005). At CW, the difference in soil thickness between eroding and non-eroding zones is less pronounced but similarly soil formation rates are faster by 0.017 mm year$^{-1}$ where soils are thinnest.

[Table 1]



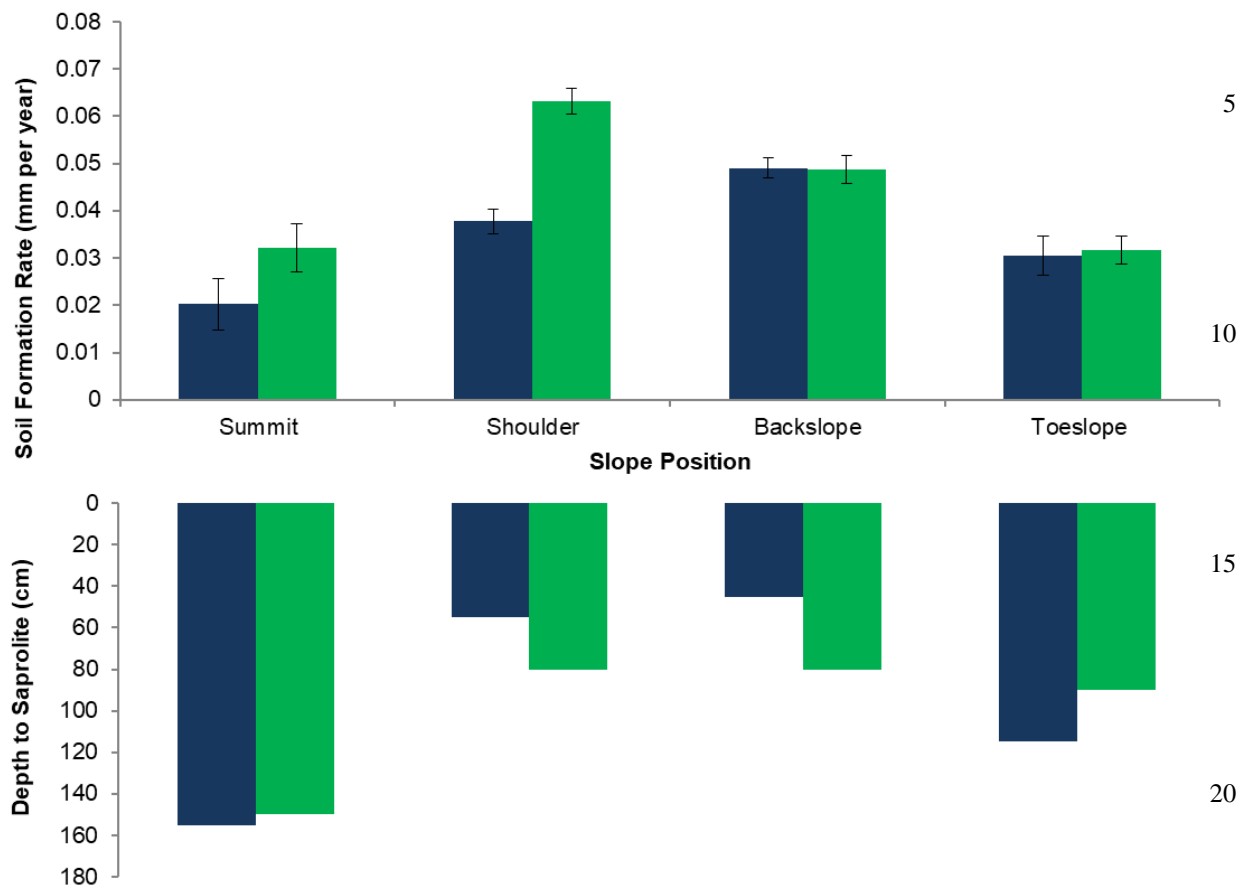

**Figure 2: Soil formation rates and the depths to saprolite for the four sampling positions along the catena transects at Rufford Forest Farm (blue) and Comer Woodland (green).**



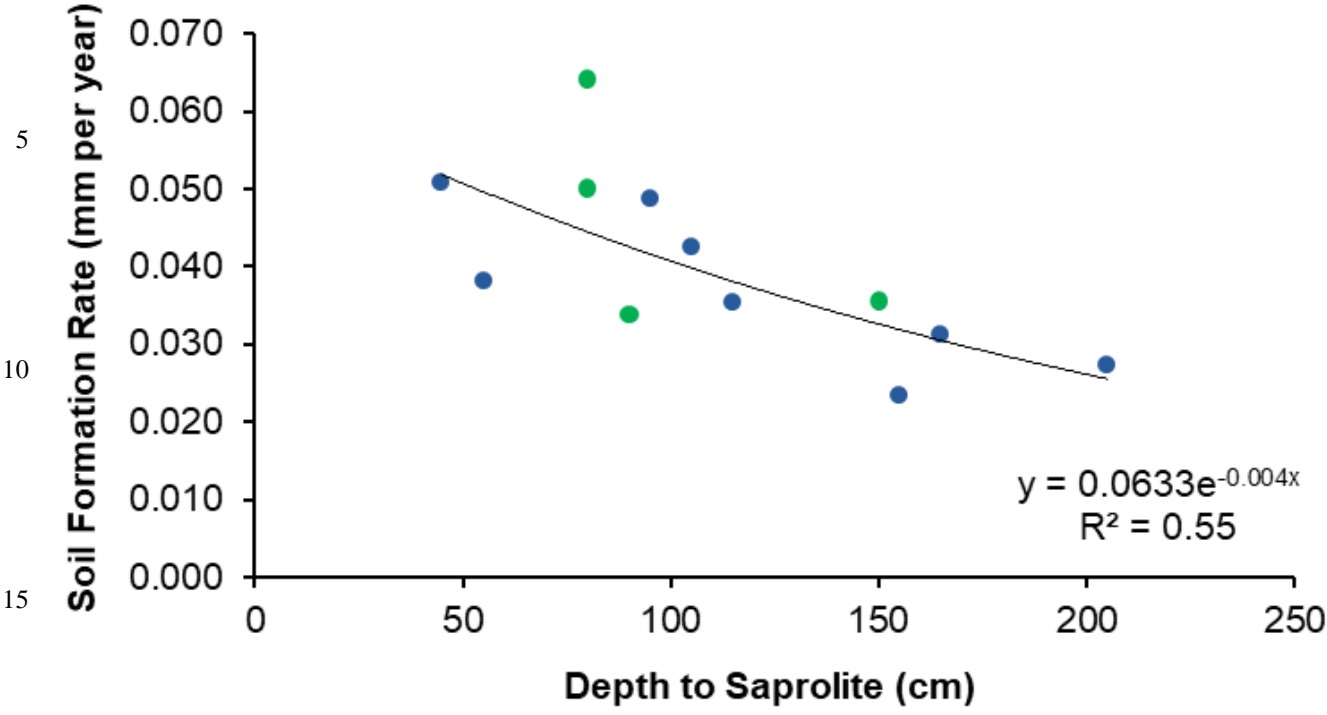

**Figure 3: Soil formation rates against the depths to saprolite for Rufford Forest Farm (blue) and Comer Woodland (green).**

Comparing data between RFF and CW demonstrates that there are other factors besides soil thickness that govern soil formation rates. For example, at the shoulder the soil thickness at CW is greater by 25 cm than that at RFF which would suggest slower formation rates. Instead soil formation rates are faster by 0.025 mm year$^{-1}$ at CW. One possible explanation is the petrographic composition of the parent material and the susceptibility of that parent material to weathering. Whilst both

RFF and CW are underlain by sandstone, the bedrock at RFF is fluvially-derived whereas that at CW is aeolian-derived. Petrological studies on fluvially-derived sandstone report a greater concentration of cementing clays in the matrix material which ultimately reduces the porosity and decreases its susceptibility to particle detachment, leading to slower soil formation rates (Wakatsuki et al., 2005; Mareschal et al., 2015).

In studies where cosmogenic methodologies have not been applied, it has been found that land use regime can promote or retard rates of bedrock weathering. Humphreys (1994) found that root channels and mesofaunal pedotubles in both the topsoil and subsoil can enhance the surface to bedrock hydrological connectivity. Similarly, Dong et al. (2018) demonstrated how an interconnected network of ecohydrologic interactions controls the supply and transport of acid to the bedrock. When a greater proportion of root mass was distributed in the uppermost horizons of the soil profile, $CO_2$ was predominantly

emitted as gas whereas when roots were distributed in the subsoil, more $CO_2$ moved downwards to increase acid production and enhance chemical weathering. Other work has sought to identify the mechanisms that affect the thermal regime of soil profiles and the consequential impacts on the weathering susceptibility of the parent material (Ahnert, 1967; Minasny and McBratney, 1999). At CW, the roots are deeper than those found observed at RFF and this is likely to proliferate weathering processes. However, given the fact that the $^{10}$Be derived soil formation rates are millennial scale averages, it is unlikely that

relatively recent (decadal-centennial) variances in the site's land use regime would be captured in the isotopic data (Darvill et al., 2013).

### 3.2 Derived soil formation rates in reference to the global inventory

Figure 4 compares soil formation rates for the study sites to an inventory of soil formation rates extracted from the published

literature (n = 252; Fig. 4a; Supplementary Table 2). The median soil formation rate in this study (0.037 mm year$^{-1}$) is 0.011 mm year$^{-1}$ faster than that of the mantled inventory but there is no statistically significant difference between the two datasets (U test; P < 0.05).  However, this global inventory comprises studies conducted on a range of geologies and climates, which are both influences on bedrock weathering rates.

Isolating the data from temperate climates (n = 187; Fig. 4b) presents a median soil formation rate of 0.035 mm year$^{-1}$, which

is 0.002 mm year$^{-1}$ slower than that measured for RFF and CW, although there is no statistically significant difference between the two datasets (U test; P < 0.05).  It is likely that the inventory's median soil formation rate for temperate climates is slower as 44% of the temperate-based data has been collected from regions that have lower mean annual precipitation than



RFF and CW which can lead to less weathering activity at the parent material (Heimsath et al., 2001; Heimsath et al., 2005; Dixon et al., 2009; Heimsath et al., 2012).

Isolating the sandstone-derived data from the inventory (n = 64; Fig. 4c) presents a median soil formation rate of 0.034 mm year$^{-1}$ which is 0.003 mm year$^{-1}$ slower than that measured for RFF and CW, although there is no statistically significant

difference (U test; P < 0.05). We suggest that faster formation rates at RFF and CW may be explained by the fact that the specific varieties of sandstone at these study sites are generally more susceptible to weathering than those within the sandstone-based inventory, of which the dominant form is the greywacke, characterised by a hard, fine-grained argillaceous matrix, with greater resistance to weathering (Cummins et al., 1962). Although there has been substantial work on the susceptibilities of major geological rock types to weathering (Stockmann et al., 2014; Wilson et al., 2017), we do not know

of any study which seeks to identify whether the susceptibility of specific varieties of sandstone have an influence on soil formation rates.

The only other study to measure soil formation rates in the UK is that of Riggins et al. (2011) where rates were derived for Bodmin Moor, Cornwall (n = 5; Fig. 4d). In that study, the median soil formation rate was 0.015 mm year-1, which is 0.022 mm year-1 slower than that for RFF and CW and statistically significant (U test; P > 0.05), despite the fact that Bodmin

Moor receives about 300 mm more precipitation per year than the sites in this study which should increase soil formation rates (Riggins et al., 2011). This is explained by the parent material at Bodmin Moor (coarse-grained granite) being generally less prone to weathering than the varieties of sandstone evident at RFF and CW (Portenga and Bierman, 2011).






**Figure 4: Soil formation rates from a globally compiled inventory (grey circles) and from this study at Rufford Forest Farm (blue
triangles) and Comer Woodland (green diamonds) plotted against soil depth. Rates in grey are from (a) the total mantled
inventory (n = 252); (b) studies from temperate climates (n = 187); (c) studies on sandstone geology (n = 64) and (d) the UK,
exclusively from Riggins et al. (2011) (n = 5). Error bars indicate the standard error.**

30

35




### 3.3 Lifespan analysis at Rufford Forest Farm

Based on a mean annual erosion rate of 1.19 mm year$^{-1}$ under arable agriculture, the lifespans of the A horizon across the catena at RFF range between 256—263 years (Figure 5). This range expands to 137—2158 years when the 5th and 95th percentile soil erosion rates are applied. However, further examination of the A horizon from cores extracted down the

catena suggest that the toeslope is in a phase of aggradation rather than thinning. Moreover, comprised within the upper stratigraphy of the soil profile down the catena is the Bunter Pebble Bed which can be found at approximately 30 cm on summit, shoulder and backslope positions but 70 cm at the toeslope. The depth to which this pebble bed occurs at the toeslope suggests that either colluviation has occurred or is still occurring, or that lifespans at this position may be either longer than 2158 years or indefinite. This demonstrates the difficulty of calculating lifespans using soil formation rates

derived from bedrock alone and not from other system inflows of soil mass such as that from colluviation and soil carbon additions.

Soil lifespans indicating the time until the exposure of the parent material span between 394—1325 years. The range of these lifespans can be explained by the fact that unlike scenario one, where a constant A horizon thickness of 30 cm was applied

across the catena, the soil thickness applied here is the depth to bedrock measured at each catena position (see Table 1). Applying upper and lower confidence intervals in the soil formation term and the 5th and 95th percentiles in the soil erosion term further widens the breadth of lifespans to 209—9394 years. The shortest lifespans are found on the backslope where bedrock exposure is expected to occur between 209—3237 years. In contrast, the greatest lifespans are found at the summit where soil thickness is 155 cm (709—9394 years). Although soil formation rates are greater at the toeslope, the depth to

bedrock is 40 cm greater at the summit and, as a result, longer durations are required for bedrock to become exposed at this position. The soil detached and transported from the backslope is expected, in part, to continue to be a contributory source of the colluvium observed at the toeslope. Although the growth of soil profiles due to colluvium is not considered in the lifespan equation, it suggests that lifespans at the toeslope may either be longer than the calculated maximum of 7372 years or indefinite.




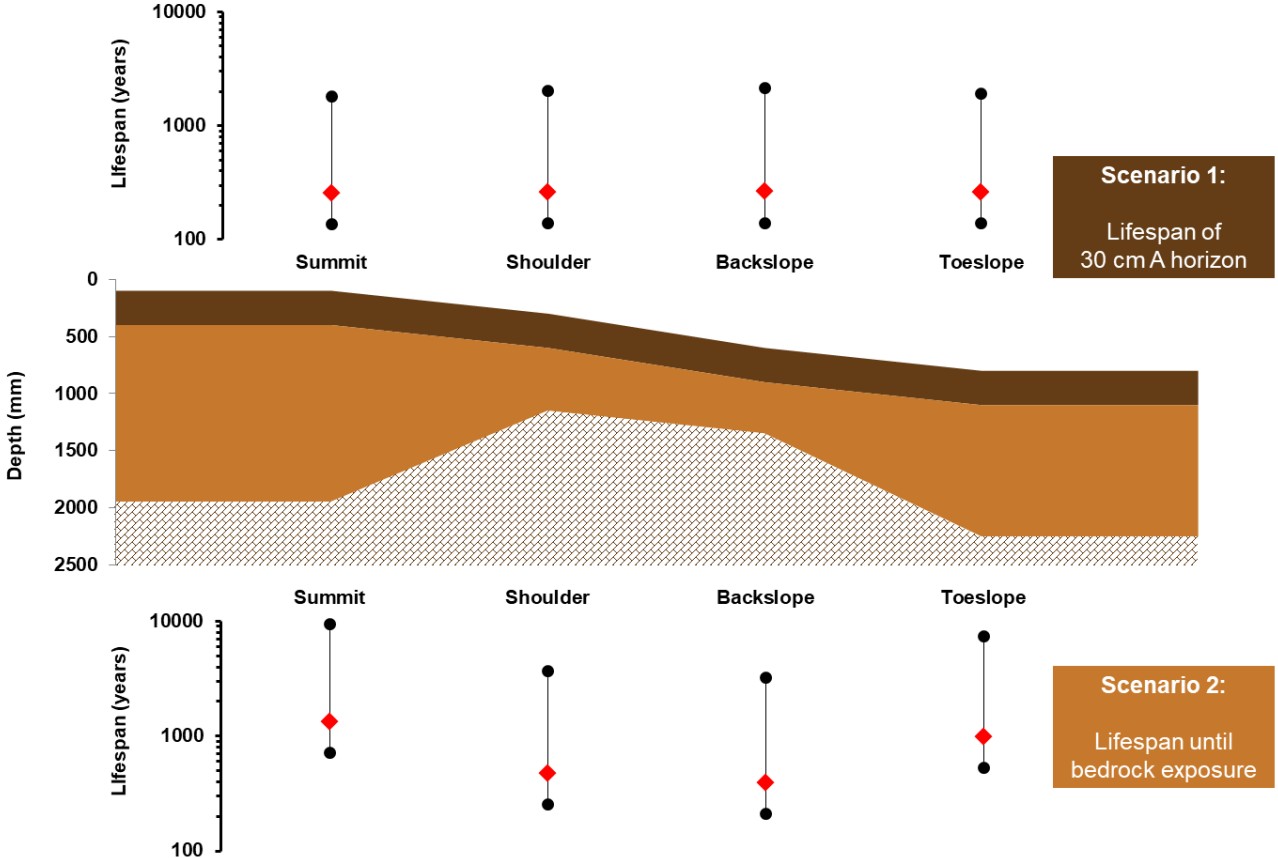

**Figure 5: First-order soil lifespans calculated at four catena positions at Rufford Forest Farm for Scenario 1 (the time until the erosion of a 30 cm A horizon) and Scenario 2 (the time until bedrock exposure). The centre diagram indicates the thickness of the A horizon (dark brown), the subsoil (light brown) and the depth to bedrock (bricks). Red diamonds denote lifespans calculated using a mean annual soil erosion rate of 1.19 mm year[-1] from Quine and Walling (1991) and soil formation rates from this study. Black dots denote the minimum and maximum lifespans calculated using the 5th and 95th percentile of the soil erosion dataset and the one sigma uncertainties in the soil formation dataset.**



The first-order lifespans presented here are based on a number of assumptions. Notwithstanding the fact that the land management regime may change within the cited time spans altering the protection the soils receive from wind and water, the erosion rates employed neither reflect the increase in the erodibility of subsoil horizons, characterised by a relatively weaker soil structure (Tanner et al., 2018) nor do they account for an expected shift in erosivity, commensurate with more

intense precipitation events (Burt et al., 2015). Acknowledging these factors, the lifespans presented here are likely to be overestimated. However, the fate of eroded soil upslope may contribute to the up-building of soil profiles in downslope concavities, extending the lifespans in the colluvial zone. In this respect the lifespans presented here, particularly those for the toeslope, are likely to be underestimated. Similarly, the soil formation rates employed.

**4.0 Conclusions**

We have presented the first isotopically-derived rates of soil formation for soils currently supporting arable agriculture. Rates derived for two UK catena sequences using cosmogenic radionuclide analysis range from $0.023 \pm 0.002$ mm year$^{-1}$ to $0.064 \pm 0.002$ mm year$^{-1}$, with mean rates being $0.037 \pm 0.003$ mm year$^{-1}$ and $0.046 \pm 0.007$ mm year$^{-1}$ for Rufford Forest Farm and Comer Wood, respectively. By combining soil formation rates from Rufford Forest Farm with soil erosion rates

derived from a prior isotopic study in a first-order lifespan model, we estimate that in a worst-case scenario the soil that currently comprises the A horizon on the backslope may be eroded in 137 years and bedrock exposure may occur in 209 years. Assessing gross soil erosion with measured rates of soil formation is important because soils that support arable agriculture are under threat from accelerated soil erosion. We have therefore shown that both the derivation and application of soil formation rates must become a fundamental component in future discussions of soil sustainability.

This work also represents the second of all isotopic studies of soil formation in the UK and therefore a significant contribution to our knowledge of pedogenesis. Soil formation rates were found to be significantly greater than those measured previously at Bodmin Moor which is explained by the fact that the parent material at Bodmin Moor is a coarse-grained granite and therefore less susceptible to weathering than the sandstone materials underlying Rufford Forest Farm and

Comer Wood. Such petrographic controls may also explain the greater rates of soil formation at Comer Wood where the sandstone matrix is largely devoid of the cementing agents present at Rufford Forest Farm and, therefore, more susceptible to particle detachment during physical and chemical weathering. Given that petrographic variability has not been thoroughly investigated in pedogenesis work, greater investment is warranted to better understand how the geochemical composition of the parent material governs the rates of soil formation.






**Data availability**

The authors declare no restrictions on the availability of materials or information.

**Supplement link**

**Author contribution**

D. L. E., J. N. Q., A. M. T. and J. A. C. D. designed research. D. L. E and A. M. T. conducted sampling. D. L. E and A. R. conducted laboratory work and analysed results. D. L. E. prepared the manuscript with contributions from all co-authors.

**Competing interests**

J. N. Q is a member of the editorial board of the journal.

**Acknowledgments**

The authors wish to thank Mr Annis (National Trust) for permission to carry out fieldwork on Comer Wood and Mr and Mrs
King (TAG Farming) for permission to carry out fieldwork on Rufford Forest Farm. We thank Vassil Karloukovski for assistance in surveying and Andrew Binley, Paul McLachlan, Jonathan Riley, Carl Horabin and the BGS Dando Drilling Rig Team for the acquisition of samples. We also wish to thank Allan Davidson, Ángel Rodés, Derek Fabel at the NERC Cosmogenic Isotope Analysis Facility for preparing samples for AMS and their subsequent assistance in data analysis. Finally, we thank Tim Quine for sharing multiple datasets from fieldwork conducted at Rufford Forest Farm. This work was
partly supported by BBSRC and NERC through a Soils Training and Research Studentships (STARS) (Grant number: NE/M009106/1) and partly by a NERC research grant (Grant number: CIAF 9179/1017). STARS is a consortium consisting of Bangor University, British Geological Survey, Centre for Ecology and Hydrology, Cranfield University, James Hutton Institute, Lancaster University, Rothamsted Research and the University of Nottingham.





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



**Table 1: $^{10}$Be concentrations and calculated maximum soil formation rates for Rufford Forest Farm (RFF) and Comer Wood (CW)**

| Site | Catena Position | Elevation, m | Horizon Position | Depth, cm | $^{10}$Be atoms, g | Uncertainty of $^{10}$Be atoms, g | $^{10}$Be production rate at surface, g$^{-1}$ year$^{-1}$ | Soil Formation Rates, mm ka$^{-1}$ | Uncertainty, mm ka$^{-1}$ |
|---|---|---|---|---|---|---|---|---|---|
| RFF | Summit | 98.7 | A | 155 | 35266 | 2364 | 4.63 | 23 | 2 |
| RFF | Summit | 98.7 | B | 205 | 22683 | 1586 | 4.63 | 27 | 2 |
| RFF | Shoulder | 99.3 | A | 55 | 54380 | 2030 | 4.63 | 38 | 1 |
| RFF | Shoulder | 99.3 | B | 105 | 30064 | 1850 | 4.63 | 43 | 3 |
| RFF | Backslope | 97.9 | A | 45 | 45603 | 1833 | 4.63 | 51 | 2 |
| RFF | Backslope | 97.9 | B | 95 | 28876 | 1661 | 4.63 | 49 | 3 |
| RFF | Toeslope | 95.7 | A | 115 | 32738 | 2006 | 4.62 | 35 | 2 |
| RFF | Toeslope | 95.7 | B | 165 | 25237 | 1562 | 4.62 | 31 | 2 |
| CW | Summit | 70.6 | A | 150 | 24507 | 1696 | 4.49 | 36 | 3 |
| CW | Shoulder | 65.3 | A | 80 | 24811 | 1333 | 4.46 | 64 | 4 |
| CW | Backslope | 58.9 | A | 80 | 31263 | 2035 | 4.42 | 50 | 3 |
| CW | Toeslope | 50.1 | A | 90 | 41276 | 1522 | 4.39 | 34 | 1 |

Horizon Position 'A' denotes the sample was taken at the soil-saprolite interface. Horizon Position 'B' denotes an additional sample was taken 50cm below the interface from the same depth profile. The shielding correction was calculated as 1.0 (to 1 d.p) for all samples and $^{10}$Be production rates are corrected for elevation and location (see Supplementary Table 1). All uncertainties are one standard deviation and are based on uncertainties in the measurement of $^{10}$Be concentration as outlined in Rodés et al. (2011). Average sample density is 2.2 g cm$^{-3}$.