# Peer review of "Arable soil formation and erosion: a hillslope-based cosmogenic-nuclide study in the United Kingdom"

_SOIL, 2019_

## Referee Comment (RC1) · Anonymous Referee #1 · 7 Jun 2019

This manuscript presents a novel and compelling investigation of soil production rates on hillslopes supporting arable agriculture. The authors pair two hillslopes with well-documented land use histories: one is under active tillage and the other, while currently forested, was open field as recently as the early 20th century. The authors carefully surveyed each slope, documenting both topography and soil morphology, and they collected a suite of samples from each hillslope for measurements of Be-10 in saprolite underlying the slopes' soil mantle. The Be-10 derived soil production rates for these sites are reasonable given their climatic and tectonic setting, and the authors do an excellent job of contextualizing their results within a global compilation of similar rates. The authors utilize previously published hillslope erosion rates derived from Cs-137 to

conduct what they refer to as a "lifespan analysis" for soils at their sites. They find a potential for complete loss of the A horizon in just over a century and the possibility for bedrock exposure in just over two centuries. In addition to being some of the first soil production rates quantified for soils supporting arable agriculture, the implications of these results are very important for future consideration of sustainable use of global soil resources.

I have now read this manuscript several times over, and I must say that I am at a loss to find any significant issues with this work. The science is sound - well-planned, well-executed, clearly/concisely described, and cleanly communicated. This paper is a pleasure to read. Each moment I found myself anticipating something they may have overlooked, the next sentence or paragraph cleared up that point. The figures are clear and adequate. The authors don't simply present their results and leave it to the reader to seek context - Figure 4 contextualizes their results simply and effectively.

If there is one place that the manuscript could be taken to the next level, it would be a more sophisticated mass and isotope balance approach to modeling hillslope soil production and transport. However, the authors are 100% transparent about the variables in their lifespan analyses, and their approach is adequate. Demanding a more detailed modeling approach does not seem an appropriate "ask" for this manuscript. I think the soil sciences and geomorphology communities will benefit most by getting these results formally published and disseminated quickly. Additional modeling can follow, if need be.

A final question that I would pose to the authors since I don't think they address it in the manuscript is this: Given that RFF has been actively farmed for over a century and a half, how do you reconcile a still extant A horizon? Do you think that tens of centimeters of soil have been lost in that time? The authors allude to the potential importance of incorporating colluvial processes at toe slopes to their work, and I would agree. Stratigraphic evidence or isotopic (Cs-137) evidence could yield some insight into the effect of the past 1.5 centuries of tillage.

In summary, it was a pleasure to review this manuscript. It is one of only a couple manuscripts that I've received for review that seem ready for publication "as is." I highly recommend this manuscript for publication in SOIL.

---

## Referee Comment (RC2) · Anonymous Referee #2 · 17 Jun 2019

General comments

This is a neat paper that provides examples of soil formation rates from bedrock weathering derived from Be10 measurements. Two catenas are presented under different current land use for soil forming from sandstone parent material. Soil lifespans are calculated for topsoil and whole soil for the arable site and are based on first order calculations that use the soil formation rates presented in this study and soil erosion estimates from previous studies at the same field location. This is a very timely study given the use in the media of the unsubstantiated quote of "There are only 60 (or 100) harvests left if soil degradation continues". A well written, concise paper that is suitable

for publication after consideration of the minor clarifications detailed below.

Specific comments

P5 line 19 Include the correct reference to WRB (2015). Please see recommended citation in the preface of the manual.

P5 L23 CW soil has 94% sand. This would classify the soil at this site in WRB as an Arenosol not a Cambisol.

P5 L19-20 Please refer to the methods used for the determination of the particle size distribution and LOI.

P7 L 5 "...observation on the competency of the extracted material..." is a bit vague - how was the Saprolite or the soil/saprolite boundary determined exactly? A change in colour, consolidation, grain size? Please provide some further details.

P7 L6 You sample at the soil-saprolite interface and 50cm below it in RFF. Please indicate the rationale for these paired samples. These samples are not differentiated in the results – so are they both used to be representative of this boundary and what are the implications for this? In table 1 the lower samples in some locations are showing active weathering indicated by greater soil formation rates.

P8 L23 An additional statement needed here to indicate the exclusion of other potential soil forming inputs (e.g. organic matter and/or aeolian dust).

P8 line 25 depth to bedrock or depth to soil/saprolite boundary? Did you only use the samples labelled A from RFF to indicate depth to saprolite? Please confirm in the text.

P13 6 Did you undertake any geochemical analysis on the samples (XRF or spectroscopy?) I guess you would have reported it but it would have been really good to see some data (perhaps in another paper...)

P13 line 27, 31; P14 L5 and 14. Check the notation for p values for the Mann-Whitney tests. For significant difference $p < 0.05$ ; for no significant difference $p > 0.05$.

P14 L3 Can you clarify if the sandstone dataset is from the temperate subset or from the whole global database? If the latter, then there is an interaction between climate and differences in sandstone lithology.

P16 L5. The toeslope also shows an Ap of 75 cm (p5 L20) which has also not been taken into account in the calculation due to the assumption that the top 30cm is representative of the current (active?) A horizon. If the top 30cm is removed then it could be argued there is still 'viable' topsoil at this location and thus the lifespan would be much greater than calculated (in addition to it also receiving colluvium).

P16 L6 Could the pebble bed offer some surface armoury that would reduce the rate of soil erosion once material above it has been eroded?

P16 line 15 This is the sampling depth, which is the soil-saprolite boundary, not depth to bedrock (be consistent with the descriptions you have used in other parts of the manuscript).

P18 L8. Is the last sentence incomplete?

Figure 2 Please indicate what the error bars show. Also include the sample numbers on the figure or in the caption.

Figure 3 If I have interpreted the sampling correctly then 4 of these samples are from 50cm below the soil-saprolite boundary. Does this figure therefore show sampling depth rather than depth to saprolite (for RFF there would be 4 pairs of samples with the same saprolite-soil boundary depth, one sample at the boundary and one 50 cm below).

Table 1 You state the average sample density. If you have measured the BD for each sample then what is the justification for using the average for all samples rather than the specific sample bulk density in the Be10 calculations?

---

## Referee Comment (RC3) · Daniel Morgan (Referee) · 19 Jun 2019

"Arable soil formation and erosion: a hillslope based cosmogenic nuclide study in the United Kingdom"

Daniel L. Evans, John N. Quinton, Andrew M. Tye, Angel Rodes, Jessica A.C. Davies, and Simon M. Mudd

Main Purpose of paper

The primary aim of this paper is to quantify soil formation rates for both agricultural land (RFF site) and for forested woodland (CW site) in the UK using the concentra-

tion of cosmogenic 10Be in saprolite from which these soils are derived. From the determined rates and previously reported soil erosion rates from agricultural land, the authors calculate soil lifespans, which can help inform policymakers and stakeholders about best land use practices.

Scientific significance

This paper contains a number of advances in our understanding of soil formation rates. 1. The second study of soil formation rates in the UK. 2. Understanding how soil formation rates vary down a hillslope transect, with varying levels of soil thickness, and for agricultural land. 3. A compilation of global soil formation rates from published papers that also utilized cosmogenic nuclides to determine soil formation rates. 4. Calculation of soil lifespans from soil formation and erosion rates.

These are significant accomplishments that warrant publication.

Scientific quality

This paper utilizes established methods, which are applied to novel sites in the UK. The methods are correctly applied and interpreted, and a sound logic flows from the structure of the paper. I think there is one small mistake in their calculations. It occurs when the authors are calculating the production rate of 10Be at the depth of their sample. In the method I propose, I get soil formation rates which are 7 to 29% higher than the authors' calculations. This does not significantly alter the main narrative, findings, or conclusions of the paper, and I think the paper is very worthy of publication.

Suggested correction for production rate calculations

Equation 1 is the correct equation to use to determine the saprolite erosion rate, which then translates into the soil formation rate. However, I would like to suggest a different way to calculate the production rate at a sample's depth (the numerator in the equation).

The authors have appropriately calculated surface production rates of cosmogenic 10Be due to spallation, fast muons, and stopping muons based on the Stone, 2000

scaling scheme. Then, to calculate the production rate of cosmogenic 10Be at the depth the samples were collected, the surface production rates are scaled with an exponential function based on the depth times the density of the overlying material. The product of depth times density is the "mass depth." In this paper, the authors appear to use the density of saprolite (2.2 g/cmˆ3) to calculate the mass depth of the samples. But the material that overlies the saprolite is soil, which should have a lower density than saprolite. I think the appropriate density to use to calculate the production rate at the sample's depth is that for soil because that represents the mass depth that overlies the soil-saprolite boundary, and the authors have (correctly) assumed that the soil thickness has not changed over time.

If one were to use the density of soil as the overlying material, instead of saprolite, the mass depth of the samples would be lower because the density of soil is lower. This would then result in a higher production rate at the depth of the samples. Then, when calculating erosion rates from equation 1, this would result in higher erosion rates because an increase in the numerator in equation 1 would require an increase in the denominator (where the erosion rate goes) to result in the same concentration of 10Be that was measured in the sample. I would like to emphasize that this impact is small, is fairly uniform across all the sample sites, and does not change the main findings of the paper.

I have recreated the authors' calculations, and performed my own calculations on the attached spreadsheet. In my experience with trying to measure soil density, soils typically have a density of 1.5 – 2.0 g/cmˆ3. In my calculations I used a value of 1.8 g/cmˆ3 as an approximate median value to my anecdotal evidence, but I would leave it to the authors to find an appropriate soil density value to use.

There are two important things to note in how I have done my calculations:

1) To calculate the mass depth, you want the depth times the density of the overlying material. For the samples at the top of the saprolite, this is simple, and is just the depth

times the density of the soil. But for the samples that are 50 cm below the top of the saprolite, this is the cumulative sum of the soil and the saprolite above the sample. This is also simple to calculate, it is the density of soil times the depth of the soil, plus the density of saprolite times 50 cm, because these samples were collected that far below the top of the saprolite.

2) This correction only applies to the numerator in equation 1. It does not apply to the denominator, which also has a depth times density term. In the case of the denominator, this is the place where erosion of the overlying material comes into the exposure model. The authors have concluded that the soil thickness does not change at a timescale that would affect the concentration of 10Be in the saprolite. I agree that this is a valid assumption, and the result is that only the saprolite changes depth with time in this exposure model. This means that only saprolite is "removed" as mass above the sample site, so the material that is eroded in equation 1 is saprolite. Thus, the density of the material in the denominator of equation 1 is correctly used at 2.2 g/cmˆ3.

The spreadsheet I have included has two tabs. The first tab on the left (Evans et al. Calculation) recreates the authors' calculation to verify that they used 2.2 g/cmˆ3 in the numerator and denominator of equation 1. The second tab contains my calculations to determine the production rate at the depth of the sample, and each corresponding new saprolite erosion rate. I've also calculated the percent difference between my calculations and those from Evans et al. Using the method I propose, the saprolite erosion rates are 7 – 29% higher than determined by Evans et al.

Although my proposed method results in higher saprolite erosion rates than those shown by the authors, the same trends discussed by the authors remain true, and the discussion and conclusions of the paper still hold. That is, the rates shown in figures 2, 3, and 5 would show the same general trends, but the numbers would be updated. Figure 4 that puts the calculated rates in context globally would have to be updated too, and that portion of the discussion could be quickly updated. Many of the tables would

need to be updated.

I suppose it's worth noting that Evans et al. have calculated the production rate of 10Be at the top of the saprolite sample. They could have made an additional correction for the sample thickness. I don't remember any discussion about sample thickness in the paper. This correction would be small, and would likely change all the numbers by only a percent or two Of course, this would depend on how thin or thick the samples were, and what the range of sample thicknesses was for the samples. I suppose that it is not necessary that they do this correction, especially if the samples are all about the same thickness and not more than a few centimeters thick. But it just occurred to me that this is missing.

Please let me finish by welcoming any discussion about my method, or that used by the authors. I think I have correctly calculated the production rate at sample depth, but I am open to discussion on the topic. If the authors think that 2.2 g/cm^3 is the correct density to use for the numerator, I would love to hear their thoughts on the question and would consider the other number.

Line by line comments:

P3, L28: You say only 252 of 1850 samples come from 10Be data. Did you compile all 1850 data points? This sounds like your compilation, and I wonder if there's more work that you've done that should be shared and part of this discussion.

P4, L31: do you mean "small" instead of "soft" when describing the grain size of the sandstone at the CW site?

P5, L3: What is the aspect of the sites? Is one north-facing and another south-facing? If you know this, it could be interesting to report as it could be a factor in the difference between the sites.

P5, L19: I don't think the citation for the FAO WRB is correctly formatted for this journal, but I'm not the expert. Is there a year?

P5, L24: I don't know what the acronym LFH stands for, and I'm not sure it's spelled out previously. If this is the first time it's used, please write it out fully.

P5, L25: This is simply a style thing, and certainly is due to my own biases. But as I read this page, I wanted to ask, "If the area had significant sediment transport from glacial outwash since the last glacial maximum, and there is a pebble layer in the stratigraphy, how certain are you that these soils are really derived from weathered saprolite?" I think the answer is, "These soils are still 82% and 94% sand, so there doesn't appear to be much input from glacial outwash into these soils." If I were writing this, I'd probably say something explicitly about this, but that's just my style and I don't think it's necessary to include this.

Also, what were the land-use practices at RFF? I thought something was written about tilling at that site, but I can't seem to find it now.

P7, L17: Equation 1 is the correct equation to use, but it does not have a time element in it. So your description of the equation above this seems a bit confusing. I think what you're missing is that once enough time has passed, the system will approach an equilibrium nuclide concentration that is the balance between the production and erosion rates. Assuming this has been reached, you can use equation 1.

You could also be more explicit that the saprolite erosion rate directly translates into the soil formation rate.

P7, L24: Were the soil pits dug and the samples collected from a vertical profile? Or were they collected from a slope perpendicular profile? Another way to put this is, was depth measured vertically or perpendicular to slope?

P8, L27: It may be worth adding a little discussion about the timescales of these measurements. The 10Be measurements represent soil formation rates that have been going on for order of 10^4 years, and the Cs-137 measurements represent erosion rates for the past ~75 years.

P9, L1: I'll admit that I'm not entirely sure why equation 2 is introduced. In this line you say you're going to derive equation 2 from the data, but you don't really ever come back to this equation with the results. I think the equation that shows up in figure 3 could be slightly altered to fit this form. It would be interesting to see something in your discussion that comes back to this equation, and the values of W and gamma that you derive, rather than seem to just assign (next note).

P9, L5: How was gamma calculated? Did it come from your data? Please elaborate. And if it came from your results, then please put it there. It is important to make your assumption that soil thickness does not impact soil production rates as sound as possible. And ultimately, you have to have that assumption to use equation 1.

P9, L15: I found the section on soil rates across the catena to be very well written and explained.

P10, Figure2: Something seems off between the graph and the data presented in Table 1. The summit of CW has a soil formation rate of 36 mm/ka in Table 1, but this appears to plot as just 30 mm/ka in Figure 2.

P13, L19: You're correct that the 10Be concentrations you measured would not be impacted by a recent landuse change, but the thickness of the soil could be changed, and this would throw off the production rate at the sample depth. As a simple example, at RFF, suppose that in the last 150 years of agriculture at the RFF site 20 cm of soil had been removed (reasonable for the Cs-137 rate, I think). The proper depth to use for the production rate would be 20 cm more than the current depth because that was the depth to the top of the saprolite for the tens of thousands of years the soil has been developing.

That is a really interesting thing to pursue. I suppose there isn't much to go on to support or negate this, but it might be worth a little bit of "error analysis" to pursue this. You could calculate the amount of soil that has been lost at RFF since agriculture started there, and include that as the steady-state soil thickness and recalculate the

production rates at the sample depths. The production rates would be lower, and the resulting soil formation rates would be lower too. You could then say something about "if we're wrong about the soil depths today being representative of the long-term soil depth, then the results would change by X percent."

P13, L26: do you want to say "soil mantled" or just mantled?

P13, L32: It would be interesting to see the data you've compiled plotted with precipitation rate. I'm also not sure I understand the discussion in this paragraph. To me, it seems like your median rate matches the median rate for the temperate climate subset. And if 44% of the temperate-based data are from regions with lower mean annual precipitation rates, that sounds like your sites are really close to the median precipitation rate of the data set. So it seems like both your precipitation rate and soil formation rate are close to this subset's median rates too. When you say there is no significant difference between the two data sets, do you mean between your results and the temperate climate subset? If so, then you do you really need to take much time explaining why you think they are different?

P14, L5: Similar to the last comment, if the data aren't statistically different, do you need to explain why you think there are differences?

P14: There does not appear to be any discussion about the results from the samples collected 50 cm below the soil-saprolite interface. These results are interesting and should be discussed. In some cases, they show faster rates than the samples from the top of the saprolite, and in other cases they are slower. In theory, they should show the same rates if soil production has been constant for a long enough time. The fact that they are different indicates that soil production hasn't been constant on the timescales these measurements record. The differences may be explained by something that has happened within the last order of 10ˆ5 years. This is because the muon attenuation lengths are much longer than that for spallation, and muons are produced at much lower rates than by spallation. The result is that muons average over much longer

timescales than spallation. Thus, when the rate in the sample 50 sample below the soil-saprolite boundary are lower than from the top of the, that may indicate that recently (order 10ˆ5 years) soil production rates increased. And vice-versa if the rate from the lower sample is higher than from the top of the saprolite. You might double-check my logic, but I think that's really cool and warrants a paragraph in this paper!

P15, Figure 4: I'm a bit confused by "depth" in this figure. Is it depth to the top of the saprolite? Or just depth below the surface? It may be helpful to know how most of the samples in this global compilation were collected. Were most from the soil-saprolite interface? Or a mix of that and below the interface like you've done?

P18: It may be appropriate to include something in your conclusion about how your results compare to the global data set you compiled.

Please also note the supplement to this comment:
https://www.soil-discuss.net/soil-2019-8/soil-2019-8-RC3-supplement.zip

---

## Author Comment (AC1) · 6 Jul 2019

Thank you for your comments on our manuscript. On behalf of my co-authors, I would like to respond to your suggestions as to how we could take this manuscript further.

(Please also refer to the PDF for equations set out below).

1a) Referee #1; C2, item 1: "If there is one place that the manuscript could be taken to the next level, it would be a more sophisticated mass and isotope balance approach to modelling hillslope soil production and transport. However, the authors are 100% transparent about the variables in their lifespan analyses, and their approach is adequate."

1b) Response to Referee #1; C2, item 1: The primary aim of the paper is to report soil formation data. The employment of these data in a first-order lifespan model is an important secondary aim and we accept that our model is currently relatively simple. To enact a more sophisticated isotope balance approach, however, we would need to execute further sampling campaigns for both sites and conduct further laboratory analyses. Whilst this is interesting work, we felt it fell beyond the scope of this paper. However, we are considering this for future work (as addressed below).

1c) Change in manuscript after Referee #1; C2, item 1: We argue that no change is necessary.

2a) Referee #1; C2, item 2: "Given that RFF has been actively farmed for over a century and a half, how do you reconcile a still extant A horizon? Do you think that tens of centimetres of soil have been lost in that time?"

2b) Response to Referee #1; C2, item 2: Soil has been redistributed downslope. This is demonstrated by the fact that the soils at the toeslope comprise, in part, of colluvium and further supported by the increased depth to the Bunter Pebble Bed at the toeslope as discussed on Page 16, line 7. Further isotopic work, particularly down the profile at the toeslope, would begin to explore this process in more detail but this is beyond the scope of this paper. There are two reasons for the survival of the extant Ap horizon. First, as soil is lost downslope, subsequent tillage operations incorporate former, unconsolidated soil from the B horizon into the Ap horizon. This leads to the dilution of the Ap horizon with the result that more of the initial Ap matrix survives than if there was no replacement. In Quine and Van Oost (2007), erosion rates are calculated from 137Cs data using the following equation: $R\_p= R\_f (1- E/P)^{ts}$ where Rf is the 137Cs fallout reference inventory, Rp is the 137Cs inventory at a point of interest, P is the cultivation layer depth, ts is the time between sampling and 1963, and E is the erosion rate. This equation can also be used to consider the survival of the Ap horizon,

where Rf is the initial Ap matrix and Rp is the surviving Ap matrix. Using the data in Quine and Van Oost (2007), Ap survival is significant eve where erosion rates of 26 t ha-1 y-1 are experienced (57% after 50 years, and 32% after 100 years). Second, we would suggest that the continuous removal of organic carbon is balanced by the dynamic replacement of new carbon input. Previous research has shown that for both water-based and tillage-based soil redistribution, this dynamic replacement rate in the upper ploughed layer exceeds that of carbon mineralisation in the sub-plough layer. (Please refer to: Van Oost et al. (2005) doi: 10.1029/2005GB002471 and papers cited therein, particularly those from Harden et al.).

2c) Change in manuscript after Referee #1; C2, item 2: We suggest that the following addition is made on Page 5, line 22: "Despite being subject to arable practices for over 150 years, the presence of a 30 cm Ap horizon may be explained in part by the incorporation of organic carbon from the B horizon, and the dynamic replacement of new carbon into the plough layer, which exceeds the rate of carbon mineralisation in the sub-plough layer (Van Oost et al., 2005) although further isotopic work is required to verify this for RFF."

3a) Referee #1; C2, item 3: "Stratigraphic evidence or isotopic (Cs-137) evidence could yield some insight into the effect of the past 1.5 centuries of tillage."

3b) Response to Referee #1; C2, item 3: We agree, and we are actively pursuing this at another site.

3c) Change in manuscript after Referee #1; C2, item 3: We argue that no change is necessary. Please note that we have signalled the need for further isotopic work within the previous 'change in manuscript' (see 2c).

Please also note the supplement to this comment:
https://www.soil-discuss.net/soil-2019-8/soil-2019-8-AC1-supplement.pdf

---

## Author Comment (AC2) · 6 Jul 2019

Thank you for your comments on our manuscript. On behalf of my co-authors, I would like to respond to your suggestions as to how we could take this manuscript further.

Please also refer to the supplementary file for the equations set out below:

1a) Referee #2; C2, item 1: "P5 line 19 Include the correct reference to WRB (2015). Please see recommended citation in the preface of the manual."

1b) Response to Referee #2; C2, item 1: We agree.

1c) Change in manuscript after Referee #2; C2, item 1: We suggest that the citation on

[Figure]

Page 5, lines 19 and 23 are changed to: "(IUSS Working Group WRB, 2015)." We will also provide a full reference in the bibliography.

2a) Referee #2; C2, item 2: "P5 L23 CW soil has 94% sand. This would classify the soil at this site in WRB as an Arenosol not a Cambisol."

2b) Response to Referee #2; C2, item 2: We agree.

2c) Change in manuscript after Referee #2; C2, item 2: We suggest that "Cambisol" is changed to "Arenosol" on Page 5, line 23.

3a) Referee #2; C2, item 3: "P5 L19-20 Please refer to the methods used for the determination of the particle size distribution and LOI".

3b) Response to Referee #2; C2, item 3: We agree.

3c) Change in manuscript after Referee #2; C2, item 3: We suggest the following addition is made on Page 8, line 9: "Soil samples were sub-sampled every 5 cm from each core at RFF and on each profile wall at CW. All samples were then oven dried overnight (105°C for 12 hours), grounded with a pestle and mortar, and sieved to discard the >2 mm fraction before being subject to particle size analysis and loss on ignition (LOI). Particle size analysis was conducted using a Beckman Coulter Laser Diffraction Particle Sizing Analyser LS 13 320 (pump speed: 70 %; sonication: 10 seconds; run-length: 30 seconds). For LOI, 5 g of each sample was placed in a Carbolite furnace CWF 1300 (550°C for 12 hours)." We then suggest that the text on Page 5, lines 19-26 ("The soils at RFF [. . .] saprolitic sandstone") are cut and are placed here on page 8, so that the description of the soil profiles follows the methods.

4a) Referee #2; C2, item 4: P7 L 5 "...observation on the competency of the extracted material..." is a bit vague - how was the Saprolite or the soil/saprolite boundary determined exactly? A change in colour, consolidation, grain size? Please provide some further details".

4b) Response to Referee #2; C2, item 4: The methods we applied both in the field and

in the laboratory were observations on the consolidation of the material supported by the penetrometer data.

4c) Change in manuscript after Referee #2; C2, item 4: We suggest a revision on Page 7, line 4: "...were later halved lengthways, and by observing the changes in the consolidation and physical integrity of the extracted material (i.e. whether it remained intact when removed from the core), together with the penetration resistance data acquired in the field, the soil-saprolite interface was demarcated." We also suggest a revision on Page 7, line 9: "Observing the changes in the consolidation and physical integrity of the material down the profile wall, together with the penetration resistance data, the soil-saprolite interface was ascertained."

5a) Referee #2; C2, item 5: "P7 L6 You sample at the soil-saprolite interface and 50cm below it in RFF. Please indicate the rationale for these paired samples. These samples are not differentiated in the results – so are they both used to be representative of this boundary and what are the implications for this? In table 1 the lower samples in some locations are showing active weathering indicated by greater soil formation rates".

5b) Response to Referee #2; C2, item 5: They are not both representative of the boundary. The first sample (labelled A in Table 1) represents the soil-saprolite interface. Since reflecting on these responses, we have made revisions to Equation 1: $N= \sum_{(i\ =\ sp,\mu\_f,\mu\hat{-})}$▒(P_i $(\theta)$ · eˆ(-x/Λ_i ))/($\lambda$+Ï\t/Λ_i )(1-eˆ(-t($\lambda$+Ï\t/Λ_i ) )) P are the annual production rates of 10Be by spallation, fast muons and stopping muons (sp, $\mu$f and $\mu$-) at a surface with slope Ï\t; x is the mass sample depth (•z); ÑĂ is the density of overburden material; z is the depth of the sample; t is the age of the landscape (the age when the original surface was generated) $\lambda$ is the decay constant of 10Be with $\lambda$ equalling In2/10Be half-life; and $\Lambda$ are the mean attenuation of cosmic radiations (Lal, 1991). t is usually considered infinite. In this paper, we tested the best fit of t based on the data from RFF. To do this, we took two samples from the same depth profile and measured the concentration of 10Be for both. At RFF, this showed that the landscape age (the time when the cosmogenic clock was reset) was >200 ka.

5c) Change in manuscript after Referee #2; C2, item 5: We suggest that Equation 1 is updated to: N= $\sum\_(i = sp,\mu\_f,\mu\hat{\ })$▒(P_i $(\theta)$ · eˆ(-x/Λ_i ))/(λ+Ïṭ/Λ_i ) (1-eˆ(-t(λ+Ïṭ/Λ_i ) ) ) ) We also suggest that a revision is made to Page 7, lines 18-23: "where: P are the annual production rates of 10Be by spallation, fast muons and stopping muons (sp, $\mu$f and $\mu$-) at a surface with slope Ïṭ; x is the mass sample depth (•z); ÑĂ is the density of overburden material; z is the depth of the sample; t is the age of the bedrock surface (the age when the original surface was generated) $\lambda$ is the decay constant of 10Be with $\lambda$ equalling In2/10Be half-life; and $\Lambda$ are the mean attenuation of cosmic radiations (Lal, 1991). t is usually considered infinite. In this paper, we took two samples from some of the sites to test if the data support this assumptions. RFF data is compatible with landscape ages >221 ka. Production rates, decay constants and attenuation lengths were calculated using field data and the CRONUS-Earth online calculator v2.3 Matlab code for the St scheme (Balco, 2008). As N can be measured using Accelerator Mass Spectrometry (AMS), Eq. (1) can be solved for $\varepsilon$ by simple interpolation of N."

6a) Referee #2; C2, item 6: "P8 L23 An additional statement needed here to indicate the exclusion of other potential soil forming inputs (e.g. organic matter and/or aeolian dust)."

6b) Response to Referee #2; C2, item 6: We agree.

6c) Change in manuscript after Referee #2; C2, item 6: We suggest the following addition is made to Page 8, line 24: "…sufficiently low, nor did we account for any allochthonous inputs to the profile such as aeolian additions and organic amendments."

7a) Referee #2; C2, item 7: "P8 line 25 depth to bedrock or depth to soil/saprolite boundary? Did you only use the samples labelled A from RFF to indicate depth to saprolite? Please confirm in the text".

7b) Response to Referee #2; C2, item 7: Yes, we used the depth to the soil-saprolite interface.

7c) Change in manuscript after Referee #2; C2, item 7: We suggest the following revision is made to Page 8, line 25: "the observed depth to the soil-saprolite interface at each catena position was employed".

8a) Referee #2; C2, item 8: "P13 6 Did you undertake any geochemical analysis on the samples (XRF or spectroscopy?) I guess you would have reported it but it would have been really good to see some data (perhaps in another paper)."

8b) Response to Referee #2; C2, item 8: We did not undertake any further analyses on the samples in this study. However, we are considering further isotopic work that may further our understanding of soil formation rates, erosion and particularly colluviation. Nevertheless, it would be beyond the primary aim of this paper to report such analysis here.

8c) Change in manuscript after Referee #2; C2, item 8: We argue that no change is necessary.

9a) Referee #2; C2, item 9: "P13 line 27, 31; P14 L5 and 14. Check the notation for p values for the Mann-Whitney tests. For significant difference $p < 0.05$ ; for no significant difference $p > 0.05$."

9b) Response to Referee #2; C2, item 9: We agree.

9c) Change in manuscript after Referee #2; C2, item 9: All reported p values should be revised (i.e: where $p < 0.05$ appears, these are replaced by $p > 0.05$, and vice versa).

10a) Referee #2; C3, item 10: "P14 L3 Can you clarify if the sandstone dataset is from the temperate subset or from the whole global database? If the latter, then there is an interaction between climate and differences in sandstone lithology."

10b) Response to Referee #2; C3, item 10: The sandstone dataset was derived from the whole global, soil-mantled database. Coincidentally, all but seven data points for the sandstone dataset stem from temperate climates (as classified by the Koppen system). The remaining seven stem from Aw (tropical/savannah) and we believe that these

should be removed from the figure so that we limit the climate signal as much as possible. (Incidentally, your point can also be made for the temperate climate dataset, reported on Page 13, line 29 onwards. The temperate climate dataset comprises rates for no less than six different parent materials).

10c) Change in manuscript after Referee #2; C3, item 10: We suggest that the seven data points not from temperate climates are removed from Figure 4c. We then suggest the addition of the following on Page 14, line 5: "Although the sandstone-derived data were derived from the global soil-mantled database, all data stem from sites in temperate climates which reduces the influence that climate may have otherwise had in this analysis on lithology." We also suggest the revision of Page 14, line 3: "(n = 57)"

11a) Referee #2; C3, item 11: "P16 L5. The toeslope also shows an Ap of 75 cm (p5 L20) which has also not been taken into account in the calculation due to the assumption that the top 30cm is representative of the current (active?) A horizon. If the top 30cm is removed then it could be argued there is still 'viable' topsoil at this location and thus the lifespan would be much greater than calculated (in addition to it also receiving colluvium)."

11b) Response to Referee #2; C3, item 11: We acknowledge that already on Page 16, line 9-10: "…lifespans at this position may be either longer than 2158 years or indefinite." We should point out that the Ap horizon of 75 cm is most likely, in part, colluvium. But we will make this clearer also on Page 8, too. Finally, we will run the lifespan model again for the toeslope and report an additional lifespan for this position, taking into account the 75 cm depth.

11c) Change in manuscript after Referee #2; C3, item 11: First, we suggest the following addition on Page 8, line 22: "…D = 30 cm across the catena. At the toeslope, an additional lifespan was calculated to account for the greater depth (75 cm) of the A horizon." Second, we suggest the following addition on Page 16, line 5: "…rather than thinning. This is supported by the fact that the depth of the Ap horizon at the toeslope is 75 cm, whereas it is 30 cm on all other observed landscape positions. Moreover, comprised within the upper stratigraphy of the soil profile down the catena is the Bunter Pebble Bed which can be found at approximately 30 cm on summit, shoulder and back-slope positions but 70 cm at the toeslope. The depth to which this pebble bed occurs at the toeslope suggests that either colluviation has occurred or is still occurring. In a scenario where colluviation is no longer active, the lifespan of this 75 cm A horizon is finite and ranges from 345 – 4808 years, but lifespans here could be longer or indefinite is colluviation continues."

12a) Referee #2; C3, item 12: "P16 L6 Could the pebble bed offer some surface armoury that would reduce the rate of soil erosion once material above it has been eroded?"

12b) Response to Referee #2; C3, item 12: An interesting idea. We discuss the potential differences in erodibility with soil removal on Page 18, line 3, but we only considered erodibility to increase. We shall acknowledge the pebble bed armoury.

12c) Change in manuscript after Referee #2; C3, item 12: We suggest the following addition on Page 18, line 3: ". . .neither reflects the increase in the erodibility of subsoil horizons, characterised by a relatively weaker soil structure (Tanner et al., 2018) nor the potential role that the Bunter Pebble Bed may play in armouring the soil surface in the future. Moreover, they do not reflect the expected shift in erosivity. . ."

13a) Referee #2; C3, item 13: "P16 line 15 this is the sampling depth, which is the soil-saprolite boundary, not depth to bedrock (be consistent with the descriptions you have used in other parts of the manuscript).

13b) Response to Referee #2; C3, item 13: We agree.

13c) Change in manuscript after Referee #2; C3, item 13: We suggest a revision to text on Page 16, line 15: "the soil thickness applied here is the depth to the soil-saprolite interface measured. . ." We also suggest a change to Figure 5 caption; ". . .(light brown)

and the depth to the soil-saprolite interface (bricks)."

14a) Referee #2; C3, item 14: P18 L8. Is the last sentence incomplete?

14b) Response to Referee #2; C3, item 14: We agree; incomplete but also superfluous.

14c) Change in manuscript after Referee #2; C3, item 14: We suggest the deletion of the final sentence on Page 18, line 8.

15a) Referee #2; C3, item 15: Figure 2 Please indicate what the error bars show. Also include the sample numbers on the figure or in the caption.

15b) Response to Referee #2; C3, item 15: We agree.

15c) Change in manuscript after Referee #2; C3, item 15: We suggest that the caption for Figure 2 includes the words: "The error bars represent one standard deviation". Further, we suggest a revision to existing text: "Rufford Forest Farm (blue; n = 4) and Comer Woodland (green; n = 4)."

16a) Referee #2; C3, item 16: Figure 3 If I have interpreted the sampling correctly then 4 of these samples are from 50cm below the soil-saprolite boundary. Does this figure therefore show sampling depth rather than depth to saprolite (for RFF there would be 4 pairs of samples with the same saprolite-soil boundary depth, one sample at the boundary and one 50 cm below).

16b) Response to Referee #2; C3, item 16: Yes, that is correct and requires an axis label change.

16c) Change in manuscript after Referee #2; C3, item 16: We suggest that the x axis label is revised to: "Sampling Depth (cm)"

17a) Referee #2; C3, item 17: Table 1 You state the average sample density. If you have measured the BD for each sample then what is the justification for using the average for all samples rather than the specific sample bulk density in the Be10 calculations?

17b) Response to Referee #2; C3, item 17: We have developed a model as part of some sensitivity analysis to be published soon. We have run the model for all sites and have incorporated the new results throughout the paper.

17c) Change in manuscript after Referee #2; C3, item 17: We suggest that results from our new analyses are incorporated throughout the paper: in Table, figures, all written analyses, etc.

Please also note the supplement to this comment:
https://www.soil-discuss.net/soil-2019-8/soil-2019-8-AC2-supplement.pdf

---

## Author Comment (AC3) · 6 Jul 2019

Thank you for your comments on our manuscript. On behalf of my co-authors, I would like to respond to your suggestions as to how we could take this manuscript further.

Please refer to the supplementary file for the equations set out below.

1a) Referee #3; C2-5, item 1: Equation 1 is the correct equation to use to determine the saprolite erosion rate, which then translates into the soil formation rate. However, I would like to suggest a different way to calculate the production rate at a sample's depth (the numerator in the equation). The authors have appropriately calculated surface production rates of cosmogenic 10Be due to spallation, fast muons, and stopping muons based on the Stone, 2000 scaling scheme. Then, to calculate the production rate of cosmogenic 10Be at the depth the samples were collected, the surface production rates are scaled with an exponential function based on the depth times the density of the overlying material. The product of depth times density is the "mass depth." In this paper, the authors appear to use the density of saprolite (2.2 g/cmËĘ3) to calculate the mass depth of the samples. But the material that overlies the saprolite is soil, which should have a lower density than saprolite. I think the appropriate density to use to calculate the production rate at the sample's depth is that for soil because that represents the mass depth that overlies the soil-saprolite boundary, and the authors have (correctly) assumed that the soil thickness has not changed over time. If one were to use the density of soil as the overlying material, instead of saprolite, the mass depth of the samples would be lower because the density of soil is lower. This would then result in a higher production rate at the depth of the samples. Then, when calculating erosion rates from equation 1, this would result in higher erosion rates because an increase in the numerator in equation 1 would require an increase in the denominator (where the erosion rate goes) to result in the same concentration of 10Be that was measured in the sample. I would like to emphasize that this impact is small, is fairly uniform across all the sample sites, and does not change the main findings of the paper. I have recreated the authors' calculations, and performed my own calculations on the attached spreadsheet. In my experience with trying to measure soil density, soils typically have a density of 1.5 – 2.0 g/cmËĘ3. In my calculations I used a value of 1.8 g/cmËĘ3 as an approximate median value to my anecdotal evidence, but I would leave it to the authors to find an appropriate soil density value to use. There are two important things to note in how I have done my calculations: 1) To calculate the mass depth, you want the depth times the density of the overlying material. For the samples at the top of the saprolite, this is simple, and is just the depth times the density of the soil. But for the samples that are 50 cm below the top of the saprolite, this is the cumulative sum of the soil and the saprolite above the sample. This is also

simple to calculate, it is the density of soil times the depth of the soil, plus the density of saprolite times 50 cm, because these samples were collected that far below the top of the saprolite. 2) This correction only applies to the numerator in equation 1. It does not apply to the denominator, which also has a depth times density term. In the case of the denominator, this is the place where erosion of the overlying material comes into the exposure model. The authors have concluded that the soil thickness does not change at a timescale that would affect the concentration of 10Be in the saprolite. I agree that this is a valid assumption, and the result is that only the saprolite changes depth with time in this exposure model. This means that only saprolite is "removed" as mass above the sample site, so the material that is eroded in equation 1 is saprolite. Thus, the density of the material in the denominator of equation 1 is correctly used at 2.2 g/cmËĘ3. The spreadsheet I have included has two tabs. The first tab on the left (Evans et al. Calculation) recreates the authors' calculation to verify that they used 2.2 g/cmËĘ3 in the numerator and denominator of equation 1. The second tab contains my calculations to determine the production rate at the depth of the sample, and each corresponding new saprolite erosion rate. I've also calculated the percent difference between my calculations and those from Evans et al. Using the method I propose, the saprolite erosion rates are 7 – 29% higher than determined by Evans et al. Although my proposed method results in higher saprolite erosion rates than those shown by the authors, the same trends discussed by the authors remain true, and the discussion and conclusions of the paper still hold. That is, the rates shown in figures 2, 3, and 5 would show the same general trends, but the numbers would be updated. Figure 4 that puts the calculated rates in context globally would have to be updated too, and that portion of the discussion could be quickly updated. Many of the tables would need to be updated. I suppose it's worth noting that Evans et al. have calculated the production rate of 10Be at the top of the saprolite sample. They could have made an additional correction for the sample thickness. I don't remember any discussion about sample thickness in the paper. This correction would be small, and would likely change all the numbers by only a percent or two Of course, this would depend on how thin or thick

the samples were, and what the range of sample thicknesses was for the samples. I suppose that it is not necessary that they do this correction, especially if the samples are all about the same thickness and not more than a few centimeters thick. But it just occurred to me that this is missing. Please let me finish by welcoming any discussion about my method, or that used by the authors. I think I have correctly calculated the production rate at sample depth, but I am open to discussion on the topic. If the authors think that 2.2 g/cmËEȝ3 is the correct density to use for the numerator, I would love to hear their thoughts on the question and would consider the other number.

1b) Response to Referee #3; C2-5, item 1: We have developed a model as part of some sensitivity analysis, using multiple bulk density measurements down the soil profile. We have assumed that the density of the overburden (soil profile) has not changed with time. (The model itself is to be published soon, elsewhere). However, we have used the model to re-calculate soil formation rates for both RFF and CW and intend to incorporate these results in the paper. We have also used the model to assess the importance of sample thickness, by calculating soil formation rates for three different sampling depths: the top, middle and bottom of the sample.

1c) Change in manuscript after Referee #3; C2-5, item 1: We suggest that results from our new analyses are incorporated throughout the paper: in Table, figures, all written analyses, etc.

2a) Referee #3; C5, item 2: "P3, L28: You say only 252 of 1850 samples come from 10Be data. Did you compile all 1850 data points? This sounds like your compilation, and I wonder if there's more work that you've done that should be shared and part of this discussion".

2b) Response to Referee #3; C5, item 2: The compilation is our work, although it is largely based off existing inventories, namely Portenga and Bierman (2011), Stockmann et al. (2014) and Montgomery (2007). These are cited on Page 3, line 27.

2c) Change in manuscript after Referee #3; C5, item 2: We argue that no change is

necessary.

3a) Referee #3; C5, item 3: "P4, L31: do you mean "small" instead of "soft" when describing the grain size of the sandstone at the CW site?"

3b) Response to Referee #3; C5, item 3: The word 'soft' here is an error.

3c) Change in manuscript after Referee #3; C5, item 3: We suggest the deletion of the word "soft" from Page 4, line 31.

4a) Referee #3; C5, item 4: P5, L3: What is the aspect of the sites? Is one north-facing and another south-facing? If you know this, it could be interesting to report as it could be a factor in the difference between the sites.

4b) Response to Referee #3; C5, item 4: Both are south-facing sites, although the effects of insolation are obviously dampened at CW due to the canopy cover.

4c) Change in manuscript after Referee #3; C5, item 4: We suggest the following addition on Page 5, line 1: "Both RFF and CW are south-facing slopes, and sit in a temperate..."

5a) Referee #3; C5, item 5: P5, L19: I don't think the citation for the FAO WRB is correctly formatted for this journal, but I'm not the expert. Is there a year?

5b) Response to Referee #3; C5, item 5: We agree.

5c) Change in manuscript after Referee #3; C5, item 5: We suggest that the citation on Page 5, lines 19 and 23 are changed to: "(IUSS Working Group WRB, 2015)." We will also provide a full reference in the bibliography.

6a) Referee #3; C6, item 6: P5, L24: I don't know what the acronym LFH stands for, and I'm not sure it's spelled out previously. If this is the first time it's used, please write it out fully.

6b) Response to Referee #3; C6, item 6: We agree.

6c) Change in manuscript after Referee #3; C6, item 6: We suggest that "LFH layer" on Page 5, line 24 is changed to "Litter Fermentation Humus layer".

7a) Referee #3; C6, item 7: P5, L25: This is simply a style thing, and certainly is due to my own biases. But as I read this page, I wanted to ask, "If the area had significant sediment transport from glacial outwash since the last glacial maximum, and there is a pebble layer in the stratigraphy, how certain are you that these soils are really derived from weathered saprolite?" I think the answer is, "These soils are still 82% and 94% sand, so there doesn't appear to be much input from glacial outwash into these soils." If I were writing this, I'd probably say something explicitly about this, but that's just my style and I don't think it's necessary to include this.

7b) Response to Referee #3; C6, item 7: We cite on Page 5, line 8 that "the prevalence of similar deposits on the study hillslope has not been studied." However, we accept that a soil with 84-94% sand suggests the absence of glacial outwash deposits; that the soils here are residual soils that have formed from sandstone rather than allochthonous sources.

7c) Change in manuscript after Referee #3; C6, item 7: We suggest the following addition to Page 5, line 25: "The sandy composition of these soils suggests that proglacial outwash deposits have not contributed to the soils of the study sites and that, instead, the soils are largely residual."

8a) Referee #3; C6, item 8: Also, what were the land-use practices at RFF? I thought something was written about tilling at that site, but I can't seem to find it now.

8b) Response to Referee #3; C6, item 8: Please refer to Page 5, line 15 where we state that "RFF has been under an arable regime and in the last twelve years, the dominant crops have been Winter Wheat and Rye." Unfortunately we are unable to provide precise details as to the tillage operations (plough depth, disc type, etc).

8c) Change in manuscript after Referee #3; C6, item 8: We argue that no change is

necessary.

9a) Referee #3; C6, item 9: "P7, L17: Equation 1 is the correct equation to use, but it does not have a time element in it. So your description of the equation above this seems a bit confusing. I think what you're missing is that once enough time has passed, the system will approach an equilibrium nuclide concentration that is the balance between the production and erosion rates. Assuming this has been reached, you can use equation 1."

9b) Response to Referee #3; C6, item 9: We agree and have now incorporated a time element, as shown in 9c.

9c) Change in manuscript after Referee #3; C6, item 9: First, we suggest the following addition is made to Page 7, line 16: "...smaller concentrations (Lal, 1991; Stockmann et al., 2014). We assume here that the production of 10Be and the erosion of the bedrock is at an equilibrium:" Second, we also suggest that Equation 1 is updated to: $N= \sum\_(i = sp,\mu\_f,\mu\hat{}-)$▒$(P\_i (\theta) \cdot e\hat{}(-x/\Lambda\_i ))/(\lambda+$Ï\ţ$/\Lambda\_i ) (1-e\hat{}(-t(\lambda+$Ï\ţ$/\Lambda\_i ) ) )$ Third, we also suggest that a revision is made to Page 7, lines 18-23: "where: P are the annual production rates of 10Be by spallation, fast muons and stopping muons (sp, $\mu$f and $\mu$-) at a surface with slope Ïţ; x is the mass sample depth (•z); ÑĂ is the density of overburden material; z is the depth of the sample; t is the age of the bedrock surface (the age when the original surface was generated) $\lambda$ is the decay constant of 10Be with $\lambda$ equalling ln2/10Be half-life; and $\Lambda$ are the mean attenuation of cosmic radiations (Lal, 1991). t is usually considered infinite. In this paper, we took two samples from some of the sites to test if the data support this assumptions. RFF data is compatible with landscape ages >221 ka. Production rates, decay constants and attenuation lengths were calculated using field data and the CRONUS-Earth online calculator v2.3 Matlab code for the St scheme (Balco, 2008). As N can be measured using Accelerator Mass Spectrometry (AMS), Eq. (1) can be solved for $\varepsilon$ by simple interpolation of N."

10a) Referee #3; C6, item 10: You could also be more explicit that the saprolite erosion

rate directly translates into the soil formation rate.

10b) Response to Referee #3; C6, item 10: For the purposes of using cosmogenic radionuclide analysis for deriving soil formation, we agree it does. However, we must (and do) consider that there are other extraneous inputs that may up-build soil profiles, which are not necessarily taken into account in a bedrock weathering rate.

10c) Change in manuscript after Referee #3; C6, item 10: We suggest the following addition is made on Page 3, line 20: "...measured and assumed to equal the rates of soil formation."

11a) Referee #3; C6, item 11: P7, L24: Were the soil pits dug and the samples collected from a vertical profile? Or were they collected from a slope perpendicular profile? Another way to put this is, was depth measured vertically or perpendicular to slope?

11b) Response to Referee #3; C6, item 11: They were vertical. We shall add this detail.

11c) Change in manuscript after Referee #3; C6, item 11: We suggest the following addition is made to Page 7, line 4: "...then proceeded to extract a series of vertical undisturbed core samples..." We also suggest that a similar addition is made to Page 7, line 8: "...a soil pit was manually dug vertically at each of the four sampling locations."

12a) Referee #3; C6, item 12: P8, L27: It may be worth adding a little discussion about the timescales of these measurements. The 10Be measurements represent soil formation rates that have been going on for order of 10Ë4 years, and the Cs-137 measurements represent erosion rates for the past 75 years.

12b) Response to Referee #3; C6, item 12: We agree.

12c) Change in manuscript after Referee #3; C6, item 12: We suggest the following addition is made to Page 8, line 30: "It should be acknowledged here that the rates of
soil formation represent timescales four orders of magnitude greater than those of soil erosion. However, if lifespans are to provide an insight into the sustainability of the soil profiles at RFF, the soil erosion rates must represent those from contemporary arable agriculture."

13a) Referee #3; C7, item 13: P9, L1: I'll admit that I'm not entirely sure why equation 2 is introduced. In this line you say you're going to derive equation 2 from the data, but you don't really ever come back to this equation with the results. I think the equation that shows up in figure 3 could be slightly altered to fit this form. It would be interesting to see something in your discussion that comes back to this equation and the values of W and gamma that you derive, rather than seem to just assign (next note).

13b) Response to Referee #3; C7, item 13: Equation 2 is introduced to test whether the rates of soil formation are sensitive to changes in soil depth, and therefore, whether a constant formation rate should be used in the denominator of Equation 3 (Page 9, line 11) or whether the formation rate should change with decreasing soil depth. It is merely a preliminary test to best formulate Equation 3. To ensure maximum transparency and accessibility, we would argue that combining this into the lifespan equation is not the best course of action.

13c) Change in manuscript after Referee #3; C7, item 13: We argue that no change is necessary.

14a) Referee #3; C7, item 14: P9, L5: How was gamma calculated? Did it come from your data? Please elaborate. And if it came from your results, then please put it there. It is important to make your assumption that soil thickness does not impact soil production rates as sound as possible. And ultimately, you have to have that assumption to use equation 1.

14b) Response to Referee #3; C7, item 14: We cite in the paper that gamma is a parameter that determines the thickness of soil when soil formation falls off by 1/e. Here, e is the exponent of the best fit exponential trend line that runs through our soil

formation rate data.

14c) Change in manuscript after Referee #3; C7, item 14: We suggest the following addition to Page 9, line 5: "The data for both the production rate (P) and the thickness of the soil (h) was used to calculate W and gamma using least squares regression."

15a) Referee #3; C7, item 15: P10, Figure2: Something seems off between the graph and the data presented in Table 1. The summit of CW has a soil formation rate of 36 mm/ka in Table 1, but this appears to plot as just 30 mm/ka in Figure 2.

15b) Response to Referee #3; C7, item 15: This is something we need to address. We have also noticed the inconsistency between Table 1 and Figure 2, and will make the necessary changes.

15c) Change in manuscript after Referee #3; C7, item 15: We shall prepare a revised figure and ensure that the data match Table 1.

16a) Referee #3; C7-8, item 16: P13, L19: You're correct that the 10Be concentrations you measured would not be impacted by a recent landuse change, but the thickness of the soil could be changed, and this would throw off the production rate at the sample depth. As a simple example, at RFF, suppose that in the last 150 years of agriculture at the RFF site 20 cm of soil had been removed (reasonable for the Cs-137 rate, I think). The proper depth to use for the production rate would be 20 cm more than the current depth because that was the depth to the top of the saprolite for the tens of thousands of years the soil has been developing. That is a really interesting thing to pursue. I suppose there isn't much to go on to support or negate this, but it might be worth a little bit of "error analysis" to pursue this. You could calculate the amount of soil that has been lost at RFF since agriculture started there, and include that as the steady-state soil thickness and recalculate the production rates at the sample depths. The production rates would be lower, and the resulting soil formation rates would be lower too. You could then say something about "if we're wrong about the soil depths today being representative of the long-term soil depth, then the results would change

by X percent."

16b) Response to Referee #3; C7-8, item 16: As part of the development of a new model to address the earlier comment on bulk density, we have also re-calculated soil formation rates assuming non-steady soil thicknesses, making use of Cs-137 derived soil erosion rates.

16c) Change in manuscript after Referee #3; C7-8, item 16: We suggest that results from our new analyses are incorporated throughout the paper: in Table, figures, all written analyses, etc.

17a) Referee #3; C8, item 17: P13, L26: do you want to say "soil mantled" or just mantled?

17b) Response to Referee #3; C8, item 17: Yes, soil mantled.

17c) Change in manuscript after Referee #3; C8, item 17: We suggest the following revision is made to Page 13, line 26: ". . .that of the soil mantled inventory. . ."

18a) Referee #3; C8, item 18: P13, L32: It would be interesting to see the data you've compiled plotted with precipitation rate. I'm also not sure I understand the discussion in this paragraph. To me, it seems like your median rate matches the median rate for the temperate climate subset. And if 44% of the temperate-based data are from regions with lower mean annual precipitation rates, that sounds like your sites are really close to the median precipitation rate of the data set. So it seems like both your precipitation rate and soil formation rate are close to this subset's median rates too. When you say there is no significant difference between the two data sets, do you mean between your results and the temperate climate subset? If so, then you do you really need to take much time explaining why you think they are different?

18b) Response to Referee #3; C8, item 18: We will add clarity to this section.

18c) Change in manuscript after Referee #3; C8, item 18: We suggest the following addition to Page 13, line 30: ". . .although there is no statistically significant difference

between those data and those we have measured for our UK study sites."

19a) Referee #3; C8, item 19: P14, L5: Similar to the last comment, if the data aren't statistically different, do you need to explain why you think there are differences?

19b) Response to Referee #3; C8, item 19: We argue that it is important to place our UK data into context. With regards to this particular example, whilst no statistical difference was found, lithological variation may still influence soil formation rates. It may be that Type 2 errors are present here.

19c) Change in manuscript after Referee #3; C8, item 19: We argue no changes are necessary.

20a) Referee #3; C8-9, item 20: P14: There does not appear to be any discussion about the results from the samples collected 50 cm below the soil-saprolite interface. These results are interesting and should be discussed. In some cases, they show faster rates than the samples from the top of the saprolite, and in other cases they are slower. In theory, they should show the same rates if soil production has been constant for a long enough time. The fact that they are different indicates that soil production hasn't been constant on the timescales these measurements record. The differences may be explained by something that has happened within the last order of 10ËE5 years. This is because the muon attenuation lengths are much longer than that for spallation, and muons are produced at much lower rates than by spallation. The result is that muons average over much longer timescales than spallation. Thus, when the rate in the sample 50 sample below the soil saprolite boundary are lower than from the top of the, that may indicate that recently (order 10ËE5 years) soil production rates increased. And vice-versa if the rate from the lower sample is higher than from the top of the saprolite. You might double-check my logic, but I think that's really cool and warrants a paragraph in this paper!

20b) Response to Referee #3; C8-9, item 20: Yes, that would be a very interesting output from the paired samples. However, our measurements are not precise enough

to solve a model with an accelerated or decelerated soil production. Actually, the calculated erosion rates from the paired samples agree within one sigma, meaning that we would not be able to prove soil formation acceleration/deceleration from these data. Also, slight changes of other factors (e.g. the actual position of the surface before farming, density uncertainties, etc.) can also affect these apparent offsets, and we have no data to rule them out.

20c) Change in manuscript after Referee #3; C8, item 20: We argue that no change is necessary.

21a) Referee #3; C9, item 21: P15, Figure 4: I'm a bit confused by "depth" in this figure. Is it depth to the top of the saprolite? Or just depth below the surface? It may be helpful to know how most of the samples in this global compilation were collected. Were most from the soil-saprolite interface? Or a mix of that and below the interface like you've done?

21b) Response to Referee #3; C9, item 21: Invariably, it is sampling depth. Many papers do not specify whether this is a depth to saprolite or to bedrock, so the best practice here is to put 'sampling depth'.

21c) Change in manuscript after Referee #3; C9, item 21: We suggest that the x axis labels are changed on all four panes to: "Sampling depth (cm)". We also suggest a revision in the caption: "...plotted against sampling depth."

22a) Referee #3; C9, item 22: P18: It may be appropriate to include something in your conclusion about how your results compare to the global data set you compiled.

22b) Response to Referee #3; C9, item 22: We agree.

22c) Change in manuscript after Referee #3; C9, item 22: We suggest that the following addition is made to Page 18, line 22: "Soil formation rates were found to fall within the range of those previously published for soils in temperate climates and on sandstone lithologies, but were found to be significantly greater than those measured

previously at Bodmin Moor. This is explained by the fact that the parent material at Bodmin Moor is a coarse-grained granite and therefore less susceptible to weathering than the sandstone materials underlying Rufford Forest Farm and Comer Wood."

Please also note the supplement to this comment:
https://www.soil-discuss.net/soil-2019-8/soil-2019-8-AC3-supplement.pdf